# Effect of a single dose of 8 mg moxidectin or 150 µg/kg ivermectin on *O. volvulus* skin microfilariae in a randomized trial: Differences between areas in the Democratic Republic of the Congo, Liberia and Ghana and impact of intensity of infection

Didier Bakajika[1¤a], Eric M. Kanza[2¤b], Nicholas O. Opoku[3¤c], Hayford M. Howard[4¤d], Germain L. Mambandu[1¤e], Amos Nyathirombo[1¤f], Maurice M. Nigo[1¤g], Kambale Kasonia Kennedy[2¤h], Safari L. Masembe[2†], Mupenzi Mumbere[2], Kambale Kataliko[2¤i], Kpehe M. Bolay[4¤j], Simon K. Attah[3¤k], George Olipoh[3¤l], Sampson Asare[3¤m], Michel Vaillant[5], Christine M. Halleux[6], Annette C. Kuesel[6]*

1 Centre de Recherche en Maladies Tropicale de l'Ituri, Hôpital Générale de Référence de Rethy, Ituri, Democratic Republic of the Congo Democratic Republic of the Congo (DRC), 2 Centre de Recherche Clinique de Butembo, Université Catholique du Graben, Site Horizon, Butembo, Nord Kivu, Democratic Republic of the Congo (DRC), 3 Onchocerciasis Chemotherapy Research Center, Hohoe, Ghana, 4 Clinical Research Center, Liberia Institute for Biomedical Research, Bolahun, Liberia, 5 Competence Center for Methodology and Statistics, Luxembourg Institute of Health, Strassen, Grand Duchy of Luxembourg, 6 UNICEF/UNDP/World Bank/WHO Special Programme for Research and Training in Tropical Diseases (WHO/TDR), World Health Organization, Geneva, Switzerland

† Deceased.
¤a Current address: ESPEN, African Regional Office of the World Health Organization (WHO/AFRO/ESPEN), Brazzaville, Republic of Congo
¤b Current address: Programme Nationale de Lutte contre les Maladies Tropicales Négligées à Chimiothérapie Préventive (PNLMTN-CTP), Ministère de la Santé Publique, Kinshasa, Democratic Republic of the Congo (DRC)
¤c Current address: Department of Epidemiology and Biostatistics School of Public Health, University of Health and Allied Sciences, Hohoe, Ghana
¤d Current address: Ganta United Methodist Hospital, Ganta City, Nimba County, Liberia
¤e Current address: Inspection Provinciale de la Santé de la Tshopo, Division Provinciale de la Santé de la Tshopo, Kisangani, Province de la Tshopo, DRC
¤f Current address: Department of Ophthalmology, Faculty of Medicine, Gulu University, Gulu, Uganda
¤g Current address: Institut Supérieur des Techniques Médicales de Nyankunde, Bunia, Ituri, DRC
¤h Current address: Department of Clinical Research, London School of Hygiene and Tropical Medicine, London, UK
¤i Current address: Centre de Santé CECA 20 de Mabakanga, Beni, Nord Kivu, DRC
¤j Current address: National Public Health Institute of Liberia, Public Health & Medical Research, Monrovia, Liberia
¤k Current address: Department of Microbiology, University of Ghana Medical School, Accra, Ghana
¤l Current address: Precious Minerals Marketing Company Ltd., National Assay Centre, Technical Department, Diamond House, Accra, Ghana
¤m Current address: GlycoScience Research Inc, Brookings, South Dakota, United States of America
* kuesela@who.int

suggestion that WHO endorses any specific organization, products or services. The use of the WHO logo is not permitted. This notice should be preserved along with the article's original URL.

**Data Availability Statement:** Participants consented to publication of summaries of the results, not to sharing of their individual data. Consequently, the Sponsor (WHO) and the authors do not have the participants' permission to make individual participant data publicly available. Individuals wanting to analyze the data should contact the Sponsor (tdr@who.int) and Medicines Development for Global Health to which WHO has licensed the data (via mark. sullivant@medicinesdevelopment.com or https:// www.medicinesdevelopment.com/contact). Requests should include the objectives, data analysis plan and plans to obtain applicable Ethics Committee approvals and involve the investigators (co-authors on this manuscript) and commitment to not share the data with anybody else. In view of the restriction to underlying data sharing, the figures and tables provided in the manuscript have been complemented with detailed output of the statistical analyses in the S1 File.

**Funding:** WHO/TDR funded this study, utilizing contributions from the WHO African Programme for Onchocerciasis Control (APOC), 6.3 million $US from Wyeth and following its acquisition by Pfizer, Pfizer, and WHO/TDR donor countries. Wyeth provided drug for this study and contributed to the study protocol. Wyeth prepared the submissions to the Ministries of Health and provided data management services until July 3, 2011. Pfizer was not further involved in this study in any way, including data verification or analysis and has not commented on this manuscript.

**Competing interests:** I have read the journal's policy and the authors of this manuscript have the following competing interests: ACK and CMH are staff of WHO which funded the work of all co-authors on the study whose data are analysed here through its department UNICEF/UNDP/World Bank/ WHO Special Programme for Research and Training in Tropical Diseases (TDR). Author Safari L Masembe was unable to confirm their authorship contributions. On their behalf, the corresponding author has reported their contributions to the best of their knowledge.

# Abstract

## Background

Our study in CDTI-naïve areas in Nord Kivu and Ituri (Democratic Republic of the Congo, DRC), Lofa County (Liberia) and Nkwanta district (Ghana) showed that a single 8 mg moxidectin dose reduced skin microfilariae density (microfilariae/mg skin, SmfD) better and for longer than a single 150μg/kg ivermectin dose. We now analysed efficacy by study area and pre-treatment SmfD (intensity of infection, IoI).

## Methodology/Principal findings

Four and three IoI categories were defined for across-study and by-study area analyses, respectively. We used a general linear model to analyse SmfD 1, 6, 12 and 18 months post-treatment, a logistic model to determine the odds of undetectable SmfD from month 1 to month 6 (UD1-6), month 12 (UD1-12) and month 18 (UD1-18), and descriptive statistics to quantitate inter-interindividual response differences. Twelve months post-treatment, treatment differences (difference in adjusted geometric mean SmfD after moxidectin and ivermectin in percentage of the adjusted geometric mean SmfD after ivermectin treatment) were 92.9%, 90.1%, 86.8% and 84.5% in Nord Kivu, Ituri, Lofa and Nkwanta, and 74.1%, 84.2%, 90.0% and 95.4% for participants with SmfD 10–20, $\geq$20-<50, $\geq$50-<80, $\geq$80, respectively. Ivermectin's efficacy was lower in Ituri and Nkwanta than Nord Kivu and Lofa (p$\leq$0.002) and moxidectin's efficacy lower in Nkwanta than Nord Kivu, Ituri and Lofa (p<0.006). Odds ratios for UD1-6, UD1-12 or UD1-18 after moxidectin versus ivermectin treatment exceeded 7.0. Suboptimal response (SmfD 12 months post-treatment >40% of pre-treatment SmfD) occurred in 0%, 0.3%, 1.6% and 3.9% of moxidectin and 12.1%, 23.7%, 10.8% and 28.0% of ivermectin treated participants in Nord Kivu, Ituri, Lofa and Nkwanta, respectively.

## Conclusions/Significance

The benefit of moxidectin vs ivermectin treatment increased with pre-treatment IoI. The possibility that parasite populations in different areas have different drug susceptibility without prior ivermectin selection pressure needs to be considered and further investigated.

## Clinical Trial Registration

Registered on 14 November 2008 in Clinicaltrials.gov (ID: NCT00790998).

## Author summary

Onchocerciasis or river blindness is a parasitic disease primarily in sub-Saharan Africa and Yemen. It can cause debilitating morbidity including severe itching, skin changes, visual impairment and even blindness. Many years of control efforts, today primarily based on mass administration of ivermectin (MDA) in endemic communities, have reduced morbidity and the percentage of infected individuals so that elimination of parasite transmission is now planned. WHO estimated that in 2020 more than 239 million people required MDA. Ivermectin may not be sufficiently efficacious to achieve

elimination everywhere. Our study in areas in Liberia, Ghana and the Democratic Republic of the Congo where MDA had not been implemented yet showed that one treatment with 8 mg moxidectin reduced parasite levels in the skin better and for longer than one treatment with 150 µg/kg ivermectin, the dose used during MDA. Here we show that people with higher numbers of parasites in the skin benefited more from moxidectin treatment than those with lower numbers and that the efficacy of ivermectin and moxidectin differed between study areas. Provided WHO and countries include moxidectin in guidelines and policies, this information could help decisions on when and where to use moxidectin.

## Introduction

Onchocerciasis is a vector-borne disease, caused by the helminth *Onchocerca volvulus*, endemic primarily in remote rural areas of sub-Saharan Africa. Different parasite life stages live in humans: the infective larvae transmitted by blackflies of the genus *Simulium* develop into macrofilariae which live primarily in subcutaneous and deep tissue nodules and produce for around 12 years millions of microfilariae which have an estimated life span of 1–2 years. The dermatological, lymphatic and ocular symptoms are due to the host inflammatory reactions to the dead microfilariae [1]. Evidence is increasing that *O. volvulus* infection is a risk factor for development of seizures and epilepsy [2].

Onchocerciasis morbidity, including severe itching, visual impairment and blindness, and its socio-economic impact have motivated large scale control programmes. Initially implemented in West Africa using vector control, these programmes are now based on mass drug administration (MDA) of ivermectin to the 'eligible population' (≥5 years or ≥90 cm, not pregnant or acutely sick) [1]. In the six Central and South American countries with an estimated 0.56 million people living in areas where the parasite was transmitted, between 23 and 36 rounds of biannual ivermectin MDA, complemented with quarterly MDA in around 300 communities, have eliminated onchocerciasis [3,4]. The exception is the endemic area across the border between Venezuela and Brazil where around 35,000 highly mobile people required MDA in 2020 [5,6]. In Sudan and Yemen, 0.79 million people required MDA in 2020 [6]. Onchocerciasis elimination faces the biggest challenge in Africa where >100 Million people live in meso- or hyperendemic areas [7–9] and more than 239 million were estimated to require MDA in 2020 [6]. In 11 countries, the Onchocerciasis Control Programme of West Africa (OCP, 1974–2002) eliminated onchocerciasis as a public health problem in the savannah areas through large scale weekly aerial larviciding of vector breeding sites along around 50,000 km of rivers, complemented later by ivermectin MDA [1]. In the other countries, the African Programme for Onchocerciasis Control (APOC, 1995–2015 [10]) facilitated ivermectin MDA implemented via community directed treatment with ivermectin (CDTI) in meso- and hyperendemic areas. Long term CDTI has significantly decreased morbidity [11,12]. For many years, it was, however, considered impossible to eliminate parasite transmission across Africa with CDTI [13]. This has motivated investments into discovery and development of more effective anti-onchocercal drugs [14]. A research study and APOC-supported evaluations of infection prevalence in areas under long term CDTI suggested that CDTI may be more effective than initially expected and could eliminate parasite transmission in many areas [15–18]. This resulted in the objective to eliminate onchocerciasis transmission in some African countries by 2020 and in 80% of endemic countries by 2025 [19,20]. These targets have recently been revised to achieve WHO-verified interruption of parasite transmission in 12 countries world-wide by 2030 [21].

Expert consultations and detailed analyses of available data spearheaded by APOC concluded that alternative treatment strategies (ATS), including those with more effective drugs than ivermectin, are needed to eliminate parasite transmission in many areas [22]. Research into different types of ATS [23] is ongoing, including discovery and development of more effective drugs [24–26], identification of effective, affordable, and sustainable complementary vector control strategies [27] and studies for implementation of new 'test-and-not-treat' or 'test-and-treat' strategies for *Loa-loa* co-endemic areas [28–32]. Furthermore, research for a vaccine is continuing [33].

The development of moxidectin, a milbemycin macrocyclic lactone, for onchocerciasis was initiated by the UNICEF/UNDP/World Bank/WHO Special Programme for Research and Training in Tropical Diseases (TDR) through pre-clinical pharmacology studies. Their results, combined with the non-clinical toxicology data acquired for registration of moxidectin for veterinary use (Document 2 in Annex 2 in [34], identified moxidectin as a clinical development candidate. Six Phase 1 studies in healthy volunteers were conducted [35–40]. A Phase 2 and a Phase 3 study in *O. volvulus* infected individuals [41,42] showed that a single oral dose of 8 mg moxidectin reduces skin microfilariae (mf) density (microfilariae/mg skin, SmfD) better and prevents repopulation of the skin with microfilariae for longer than a single oral dose of ivermectin as used during CDTI (150 μg/kg). In the Phase 3 study, the treatment difference (calculated as the difference between the adjusted geometric mean SmfD in the two treatment arms in percentage of the adjusted geometric mean SmfD in the ivermectin treatment arm) was 86%, 97%, 86% and 76% at 1, 6, 12 and 18 months after treatment (p<0.0001). The treatment difference was independent of sex, but higher for individuals with ≥20 SmfD (93% at month 12) than for individuals with <20 SmfD (76.0% at month 12) pre-treatment [42]. Both studies showed that moxidectin and ivermectin have comparable safety profiles. In conjunction with the safety data from the healthy volunteer studies and non-clinical toxicology studies, the Phase 2 and Phase 3 study safety data suggest that moxidectin may have the safety profile required for Community-Directed Treatment with Moxidectin (CDTM). The Food and Drug Administration of the United States of America (US FDA) approved moxidectin for the treatment of onchocerciasis in ≥12-year-olds on 13 June 2018 following priority review of a New Drug Application submitted by Medicines Development for Global Health (MDGH). MDGH is the Australian not-for-profit biopharmaceutical company to whom WHO had licensed all moxidectin-related data at its disposal to register moxidectin. [43]

The previous report of the Phase 3 study [42] analyzed the efficacy of moxidectin and ivermectin across all participants and by the two pre-treatment SmfD categories used for stratifying participants at randomization. Here, we are reporting analyses conducted to investigate whether the efficacy of the drugs differs between study areas, to better characterize the dependency of the treatment difference on pre-treatment SmfD and to explore the significant inter-individual variability of response to ivermectin the raw data suggested [42]. Furthermore, we are reporting the *O. volvulus* infection relevant data obtained in all individuals screened by study area.

## Methods

Details for all methods except the statistical analyses for this manuscript have been provided previously [42] and are only summarized briefly.

### Ethics statement

Conduct of this study, including protocol, information documents for potential participants and the participant consent forms, were approved by the Ghana Food and Drugs Authority

and the Ghana Health Service Ethics Review Committee, the Liberia Ministry of Health and Social Welfare and the Ethics Committee of the Liberia Institute for Biomedical Research, the Ministère de la Santé Publique of DRC and the Ethics Committee of the Ecole de la Santé Publique Université de Kinshasa in DRC, and the WHO Ethics Review Committee.

Volunteers provided their consent or assent with parental consent to study participation through signature or thumbprint in the presence of a literate witness in their villages. This included consent to publication of summaries of the results.

## Trial registration

The study was registered on 14 November 2008 in Clinicaltrials.gov (ID: NCT00790998).

## Study design and objectives

The double-blind, randomized, ivermectin-controlled study in *O. volvulus* infected individuals was designed to determine whether a single 8 mg oral dose of moxidectin achieves SmfD ≤50% of SmfD after a single oral dose of 150μg/kg ivermectin one year after treatment and to collect safety data.

## Study timing and areas

The study was conducted from 2009 to 2012 in four onchocerciasis endemic areas in which CDTI had not yet been initiated: Nord Kivu province (current Zones de Santé Kalunguta and Mabalako) in the Democratic Republic of the Congo (DRC), Ituri province in DRC (Zone de Santé Logo in Northern Ituri, subsequently referred to as Nord Ituri), Lofa county in Liberia (subsequently referred to as Lofa or Liberia) and the Kpasa subdistrict within the Nkwanta North health district in Ghana (subsequently referred to as Nkwanta or Ghana).

## Participant eligibility criteria

Volunteers had to be ≥12 years old, weigh ≥30 kg and have ≥10 SmfD. Women of reproductive capacity could not be pregnant and had to commit to using reliable contraception (abstinence, condoms or hormonal contraception as medically recognized methods) for 6 months after treatment (for ivermectin and moxidectin labelling regarding use in pregnant women see, respectively, https://www.merck.com/product/usa/pi_circulars/s/stromectol/stromectol_pi.pdf and https://www.accessdata.fda.gov/scripts/cder/daf/index.cfm?event=overview.process&varApplNo=210867). To include a population as similar to that which would be participating in CDTM as possible, volunteers meeting inclusion criteria consistent with available pre-clinical and clinical safety data were excluded only if they met criteria potentially compromising their safety, efficacy assessments or safety assessments.

## Randomization and treatment

Eligible participants were randomized by study area, sex and pre-treatment SmfD (<20 vs. ≥ 20 mf/mg skin) in a ratio of 2:1 to 8 mg moxidectin or 150 μg/kg ivermectin using computer-generated randomization lists. To ensure double blinding, a pharmacist not involved in participant evaluation provided for each participant four identical looking capsules containing 2 mg moxidectin tablets, 3 mg ivermectin tablets or placebo, as required by treatment allocation and weight. A study team member observed participants while they were swallowing the capsules.

## Measurement of skin microfilariae densities

Four skin snips were obtained (one snip from each iliac crest and calf) pre-treatment and 1, 6, 12 and 18 months post-treatment and SmfD (microfilariae/mg skin) determined as described previously [41,42].

## Duration of follow up

The study was initiated with last follow up occurring 18 months post-treatment. From July 2011 onward, WHO was the sole study sponsor. Resulting resource limitations required a protocol amendment eliminating the 18 months follow up examinations. This resulted in 96% of moxidectin treated and 97% of ivermectin treated participants with 12 months follow up but only 78% of participants in both treatment arms with month 18 follow up data [42].

## Sample size

The sample size required to show a ≥50% difference in the primary efficacy outcome SmfD 12 month after treatment with $\alpha = 0.05$ and 90% power and the 2:1 (moxidectin:ivermectin) randomization ratio was 185. For a better characterization of the safety profile, the planned sample size was 1000 moxidectin and 500 ivermectin treated individuals.

## Statistical analysis

In analogy to the Community Microfilarial Load [44], the microfilarial load among screened individuals (ScMFL) was calculated as the geometric mean of (SmfD+1) -1 for all villages in whom at least 18 individuals ≥20 years old were screened.

For analysis by pre-treatment SmfD, four SmfD categories, referred to as 'intensity of infection' (IoI) categories, were defined for analyses across study areas: 10 to <20, 20 to < 50, 50 to < 80 and ≥ 80. Considering the number of participants with ≥80 pre-treatment SmfD, only three IoI categories (10 to <20, 20 to < 50, ≥ 50) were defined for analyses by study area.

For inferential statistical analysis, SmfD were log-transformed ($y = \log_e(SmfD+1)$) before analysis.

Longitudinal analysis of SmfD 1, 6, 12 and 18 months after treatment used a general linear model for repeated measures with pre-treatment SmfD, treatment, sex, IoI category, time, treatment*IoI category, and treatment*time interaction as fixed effects, time as repeated effect for the four SmfD measurements with participant as the statistical unit. In the across study area analysis, study area was a random effect (for sensitivity analysis regarding study area as fixed or random effect see [42]). The treatment difference was calculated as the difference in adjusted geometric means (obtained from the model) of the two treatment arms in percentage of the adjusted geometric means in the ivermectin arm. Pairwise comparison of the efficacy of ivermectin and of moxidectin between the four study areas was conducted through marginal means comparisons (least squares means from the model). All participants who received study drug were included in these analyses (modified Intent to Treat population, mITT).

The odds of participants having undetectable SmfD at month 1 and 6 (UD1-6), at month 1, 6 and 12 (UD1-12) and at month 1, 6, 12 and 18 (UD1-18) after treatment with moxidectin or ivermectin were calculated using a logistic model with treatment, lol category and IoI category*treatment interaction as fixed effects. For the across study area analysis, the study area was a random effect. Only individuals who had SmfD determined at each of the relevant time points were included in these analyses. Note that the statistical analysis plan signed off prior to unblinding planned on analysis of percentage of participants with undetectable SmfD by

measurement time point as presented in [42]. The analysis of UD 1–6, UD1-12 and UD1-18 was added.

Inter-individual differences in response to ivermectin and moxidectin between and within study areas and the extent to which the response was a function of the pre-treatment SmfD, was explored with descriptive analyses by study area and IoI category. The measures to describe inter-individual differences used were: (1) 'suboptimal microfilariae response' (SOMR, reduction of ≤ 80% in SmfD from pre-treatment to month 1 (SOMR80) and reduction of ≤ 90% in SmfD from pre-treatment to month 1 (SOMR90)), (2) detectable SmfD at all post-treatment measurements to Month 12 and minimum SmfD, and (3) 'suboptimal response' (SOR, SmfD at month 12 post treatment of >40% of the pre-treatment SmfD [45]).

Descriptive statistics and figures were generated with STATA 13.1 (StataCorp LLC, USA) or Excel 2016. Inferential statistical analyses were conducted with SAS System version 9.4 (SAS Institute, Cary, NC, USA).

## Results

### Consort diagram

The CONSORT flow diagram has been reported previously [42].

### Prevalence and intensity of *O. volvulus* infection among screened individuals

The demographic and *O. volvulus* infection characteristics of all individuals screened are provided by study area in Tables S1-S3 and Figs S1-S3 in **S1 File**.

Even taking into account that (a) no consideration was given to choosing villages for recruitment based on being first or higher line villages and that (b) those volunteering for screening were not randomly selected and may be biased towards those suspecting themselves to be infected, the screening data show that in each study area, onchocerciasis was meso- or hyperendemic.

### Demographic and pre-treatment *O. volvulus* infection characteristics of enrolled study participants by study area

Table 1 shows the demographic and pre-treatment characteristics of enrolled study participants by study area and treatment arm.

### Effect of moxidectin and ivermectin on skin microfilariae density

The number of individuals treated and with follow up data as well as descriptive statistics of SmfD by IoI category across all study areas and by study area are available in Tables S4-S8 in **S1 File**. Across the study, follow-up rates in the two treatment arms were >99% at month 1, 98–99% at month 6, 96–97% at month 12 and, due to the protocol amendment, 78% at month 18 [42].

### Post treatment skin microfilariae densities by study area

Fig 1 shows the SmfD before and 1, 6, 12 and 18 months post treatment by study area. The percentage reduction in geometric mean SmfD at baseline to month 1 ranged from 99.7%-100% in the moxidectin and from 97.4%-97.9% in the ivermectin treatment arm. Table 2 shows the treatment difference between the moxidectin and ivermectin treatment arm at each post-treatment evaluation by study area.

**Table 1. Demographics and intensity of infection of enrolled study participants by study area.**

| Study area | Parameter (unit) | Statistic | Moxidectin | Ivermectin |
|---|---|---|---|---|
| **Nord Kivu** | Women | N (%) | 112 (36.7) | 58 (37.4) |
| | Men | N (%) | 193 (63.3) | 97 (62.6) |
| | Age (yrs) | Mean±SD | 45.5±14.1 | 47.9±14.7 |
| | Age (yrs) | Min, Max | 16, 82 | 13, 86 |
| | Height (cm) | Mean±SD | 155.9±7.95 | 156.0±8.35 |
| | Weight (kg) | Mean±SD | 49.8±7.79 | 49.2±8.06 |
| | IoI (mf/mg skin) | Mean±SD | 34.4±23.36 | 39.1±28.84 |
| | IoI 10-<20 (mf/mg skin) | N (%) | 92 (30.2) | 51 (32.9) |
| | IoI 20-<50 (mf/mg skin) | N (%) | 151 (49.5) | 53 (34.2) |
| | IoI 50-<80 (mf/mg skin) | N (%) | 48 (15.7) | 42 (27.1) |
| | IoI ≥80 (mf/mg skin) | N (%) | 14 (4.6) | 9 (5.8) |
| **Nord Ituri** | Women | N | 115 (36.5) | 57 (36.3) |
| | Men | N | 200 (63.5) | 100 (64.7) |
| | Age (yrs) | Mean±SD | 39.6±15.12 | 41.7±14.29 |
| | Age (yrs) | Min, Max | 12, 74 | 15, 81 |
| | Height (cm) | Mean±SD | 159.0±8.50 | 160.7±7.52 |
| | Weight (kg) | Mean±SD | 52.0±8.00 | 53.3±6.95 |
| | IoI (mf/mg skin) | Mean±SD | 48.1±39.23 | 52.6±38.36 |
| | IoI 10-<20 (mf/mg skin) | N (%) | 64 (20.3) | 33 (21.0) |
| | IoI 20-<50 (mf/mg skin) | N (%) | 145 (46.0) | 53 (33.8) |
| | IoI 50-<80 (mf/mg skin) | N (%) | 57 (18.1) | 37 (23.6) |
| | IoI ≥80 (mf/mg skin) | N (%) | 49 (15.6) | 34 (21.7) |
| **Lofa** | Women | N (%) | 80 (40.0) | 40 (40.4) |
| | Men | N (%) | 120 (60.0) | 59 (59.6) |
| | Age (yrs) | Mean±SD | 48.0±17.47 | 46.9±16.49 |
| | Age (yrs) | Min, Max | 19, 95 | 18, 76 |
| | Height (cm) | Mean±SD | 161.3±8.03 | 162.5±7.71 |
| | Weight (kg) | Mean±SD | 53.2±7.83 | 53.6±6.83 |
| | IoI (mf/mg skin) | Mean±SD | 30.7±22.33 | 29.3±19.53 |
| | IoI 10-<20 (mf/mg skin) | N (%) | 83 (41.5) | 42 (42.4) |
| | IoI 20-<50 (mf/mg skin) | N (%) | 84 (42.0) | 39 (39.4) |
| | IoI 50-<80 (mf/mg skin) | N (%) | 26 (13.0) | 15 (15.2) |
| | IoI ≥80 (mf/mg skin) | N (%) | 7 (3.5) | 3 (3.0) |
| **Nkwanta** | Women | N (%) | 45 (28.5) | 24 (28.9) |
| | Men | N (%) | 113 (71.5) | 59 (71.1) |
| | Age (yrs) | Mean±SD | 29.4±14.42 | 30.6±14.85 |
| | Age (yrs) | Min, Max | 12, 65 | 12, 64 |
| | Height (cm) | Mean±SD | 161.7±9.69 | 159.8±9.58 |
| | Weight (kg) | Mean±SD | 52.2±10.26 | 50.5±9.15 |
| | IoI (mf/mg skin) | Mean±SD | 39.2±27.25 | 37.5±25.08 |
| | IoI 10-<20 (mf/mg skin) | N (%) | 42 (26.6) | 24 (28.9) |
| | IoI 20-<50 (mf/mg skin) | N (%) | 76 (48.1) | 38 (45.8) |
| | IoI 50-<80 (mf/mg skin) | N (%) | 26 (16.5) | 14 (16.9) |
| | IoI ≥80 (mf/mg skin) | N (%) | 14 (8.9) | 7 (8.4) |

IoI intensity of infection (pre-treatment skin microfilariae density), mf microfilariae

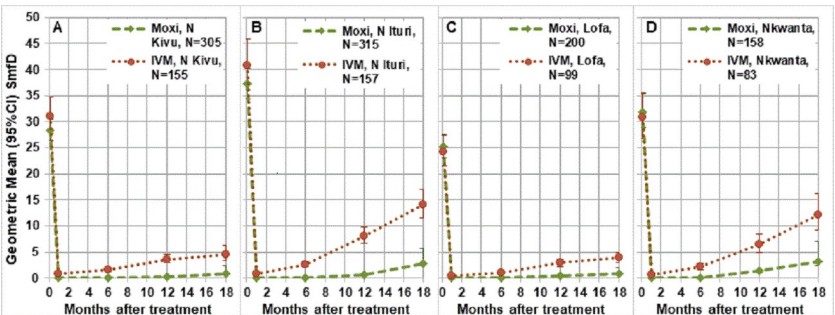

**Fig 1. Geometric mean (95% CI) SmfD before and after treatment by study area.** N Kivu Nord Kivu, N Ituri Nord Ituri, Lofa Lofa County, Nkwanta Nkwanta North district, N number treated (for number with follow up data at each time point, see Tables S5-S8 in **S1 File**).

SmfD increased from month 1 onward in the ivermectin arm and from month 6 onward in the moxidectin treatment arm. These increases were higher in Nord Ituri and Nkwanta North than in Nord Kivu and Lofa County. The similar mean IoI for Nord Kivu and Nkwanta North and similar percentage of participants with ≥50 mf/mg skin IoI suggest that the differences in SmfD increases between the study areas is unlikely to be due solely to differences in IoI among participants (Fig 1). Pairwise study area comparison of the SmfD 12 months after treatment with adjustment for IoI showed that the efficacy of IVM was similar in Nord Kivu and Lofa County as well as in Nord Ituri and Nkwanta district whereas the efficacy in Nord Kivu and Lofa County was significantly higher than that in Nord Ituri and the Nkwanta district (p≤0.0002). The efficacy of moxidectin was similar in Nord Kivu, Nord Ituri and Lofa County and significantly higher in these three areas than in Nkwanta district (p<0.006).

## Post-treatment skin microfilariae densities by pre-treatment intensity of infection

The difference in SmfD post-treatment between the moxidectin and ivermectin treatment arm increased with increasing IoI (Fig 2). Twelve months post-treatment (i.e. the time of the next treatment during an annual moxidectin treatment strategy), the treatment difference across study areas was 74.1%, 84.2%, 90.0% and 95.4% among those with IoI of 10-<20, ≥20-<50, ≥50-<80, ≥80, respectively (p<0.0001). In the linear model, both the interaction of treatment and time and the interaction of treatment and IoI category confirmed (p<0.0001) that the difference in SmfD post-treatment between the moxidectin and ivermectin treatment arms increased over time as well as with IoI categories.

**Table 2. Treatment difference 1, 6, 12 and 18 months post-treatment by study area.**

| Time post treatment | DRC Nord Kivu | DRC Nord Ituri | Liberia Lofa | Ghana Nkwanta |
|---|---|---|---|---|
| Month 1 | 98.6% | 83.7% | 97.1% | 110.7% |
| Month 6 | 100.4% | 98.8% | 98.8% | 103.2% |
| Month 12 | 92.9% | 90.1% | 86.8% | 84.5% |
| Month 18 | 84.1% | 78.0% | 78.2% | 78.2% |

Treatment difference: difference between adjusted geometric means in the ivermectin treatment arm and the moxidectin treatment arm in percentage of the adjusted geometric means in the ivermectin treatment arm obtained from the linear model. Treatment differences >100% are due to low values of SmfD in the moxidectin arm that are log-transformed to calculate geometric mean differences. For summaries of skin microfilariae density means, geometric means, adjusted means and geometric means and adjusted difference of means and geometric means see Tables S5-S8 in **S1 File**.

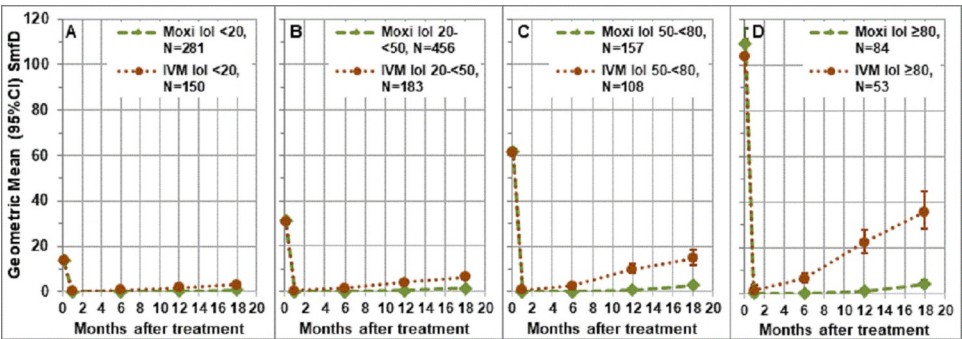

**Fig 2. Geometric mean (95% CI) SmfD by intensity of infection category across study areas.** SmfD skin microfilariae density, moxi moxidectin, IVM ivermectin, N number treated (for number with follow up data at each time point, see Table S4 in **S1 File**).

The analysis by study area showed the same trend towards higher treatment differences among participants in the higher IoI categories as the across study area analysis. In all study areas, participants in the highest IoI category benefitted more from moxidectin treatment than those in the lowest IoI category (Fig 3 and Table 3). Descriptive statistics for the SmfD by IoI category across and by study area are provided in Tables S4-S8 in S1 File.

## Percentage of participants with undetectable SmfD post treatment by study area and pre-treatment intensity of infection

Across all study areas, 151/959 (15.7%) of participants with data at both 1 and 6 months after moxidectin treatment had lower SmfD 6 months (0.02±0.11 mf/mg, range 0–1.23 mf/mg) than 1 month (0.53±1.03, range 0.05–8.37 mf/mg) after treatment resulting in 136 participants achieving undetectable SmfD only at 6 months. Since a high percentage of moxidectin treated participants retained undetectable SmfD from month 1 to month 6, this resulted in a higher percentage of participants with undetectable SmfD from Month 1 to Month 6 (Fig 4) than at month 1 (84%, [42]). Such an increase was not observed in the ivermectin treatment arm, but 107/490 (21.8%) of participants in the ivermectin treatment arm with data at both 1 and 6 months had lower SmfD 6 months (4.50±8.66 mf/mg, range 0–43.1 mf/mg) than 1 month (7.81±13.36 mf/mg, range 0.08–53.6 mf/mg) after treatment.

It is noteworthy that small changes in SmfD from one follow up time point to the other may not reflect actual changes in SmfD but rather the accuracy of SmfD detection. This has a relatively low impact on analyses based on mean SmfD, but can have a significant impact on categorical analyses such as the percentage with undetectable SmfD. The majority of moxidectin treated participants with detectable SmfD at Month 1 and undetectable SmfD at Month 6 had <0.25 mf/mg skin at Month 1 (Fig 5).

## Duration of undetectable levels of SmfD

The number of individuals in each study area with and without undetectable SmfD and those not evaluable for this analysis because SmfD data were missing for at least one of the relevant time points is provided in Table S11 in S1 File. The logistic model derived odds and their lower and upper limits are provided in Table S12 in S1 File and displayed in Fig S4 in S1 File and the odds ratios are provided in Table S13 in S1 File. Table S11 in S1 File shows that there were zero participants with undetectable SmfD in the ivermectin arm in the higher IoI categories which resulted in the logistic model not providing odds and odds ratios (Tables S12 and S13 in S1 File).

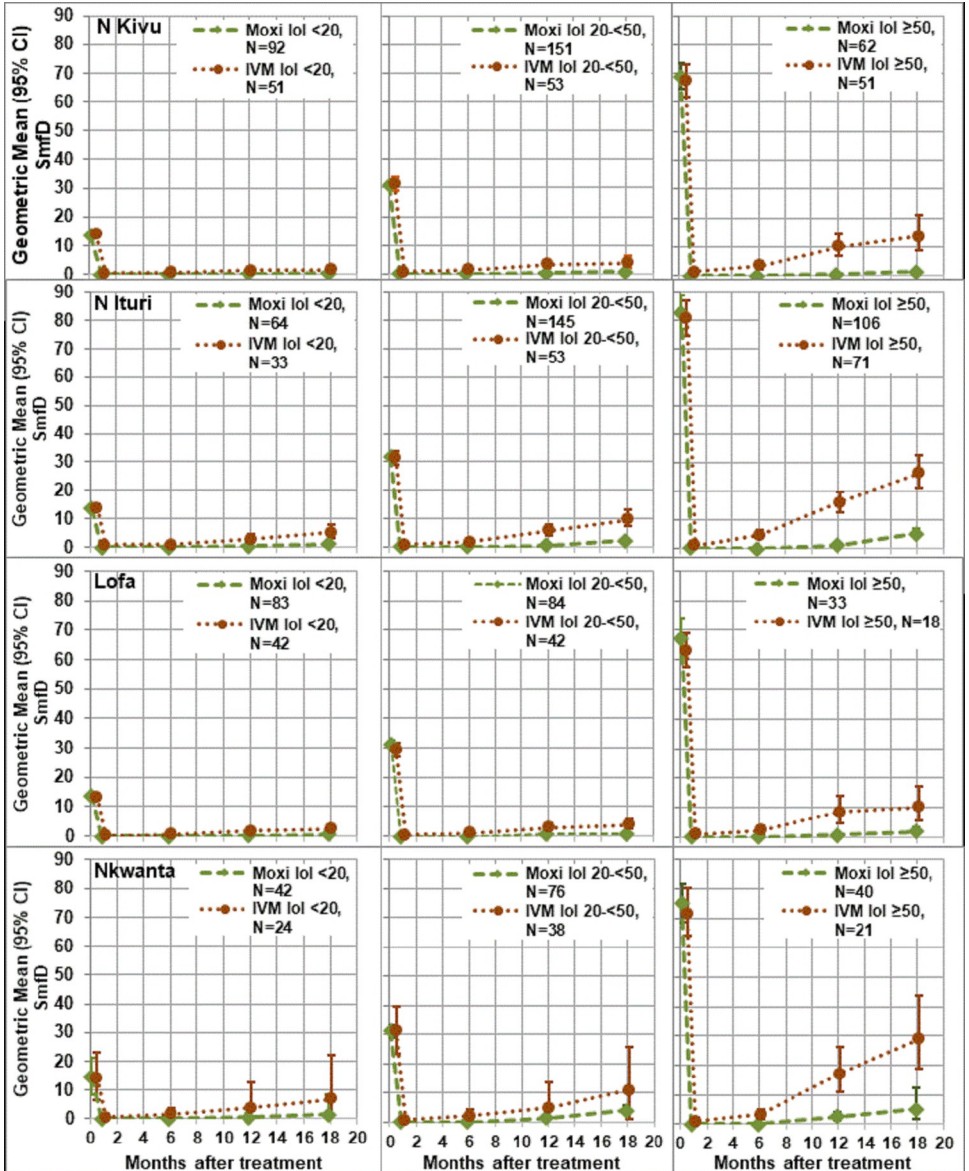

**Fig 3. Geometric mean (95% CI) SmfD by intensity of infection category and study area.** Moxi: moxidectin, IVM: ivermectin, N: number treated (for number with follow up data at each time point, see Tables S4-S8 in **S1 File**), IoI intensity of infection: <20: 10-<20 mf/mg skin; 20–50: 20-<50 mf/mg skin, ≥50: ≥50 mf/mg skin pre-treatment.

Fig 6 shows the logistic model derived odds of study participants across all study areas to have UD from 1–6, 1–12 and 1–18 months after treatment with moxidectin or ivermectin by the four IoI categories defined for the across study area analysis.

The odds of moxidectin treated participants to have UD of any duration were significantly higher than those of ivermectin treated participants in each IoI category (p<0.0002 for each IoI category) but the difference in odds between the moxidectin and ivermectin arms was not significantly different between IoI categories (p>0.8 for the interaction between treatment and IoI which was discarded from the models). The highest odds in the ivermectin treatment arm were 0.13 for individuals with <20 mf/mg IoI to have UD 1–6.

**Table 3. Treatment difference 12 months post-treatment by study area across and by intensity of infection.**

| Pre-treatment SmfD | DRC Nord Kivu | DRC Nord Ituri | Liberia Lofa | Ghana Nkwanta |
|---|---|---|---|---|
| (Any) ≥10 mf/mg | 92.9% | 90.1% | 86.8% | 84.5% |
| 10 - <20 mf/mg | 68.1% | 75.7% | 81.2% | 83.9% |
| ≥20 - <50 mf/mg | 89.6% | 83.9% | 85.3% | 80.2% |
| ≥ 50 mf/mg | 101.6% | 89.9% | 87.9% | 88.2% |

The percentage treatment difference was calculated as the difference in adjusted geometric means between treatment arms in percentage of the post-ivermectin adjusted geometric mean obtained from a linear model for repeated measures with pre-treatment IoI, treatment, IoI category, time, treatment*IoI category, and treatment*time interaction as fixed effects, time as repeated effect for the four SmfD measurements with participant as statistical unit. p<0.0001 for all treatment differences. Treatment difference >100% is due to low values of SmfD in the moxidectin arm that are log-transformed to calculate geometric mean differences. For adjusted means and geometric means and adjusted difference of means and geometric means see Tables S9 and S10 in **S1 File**.

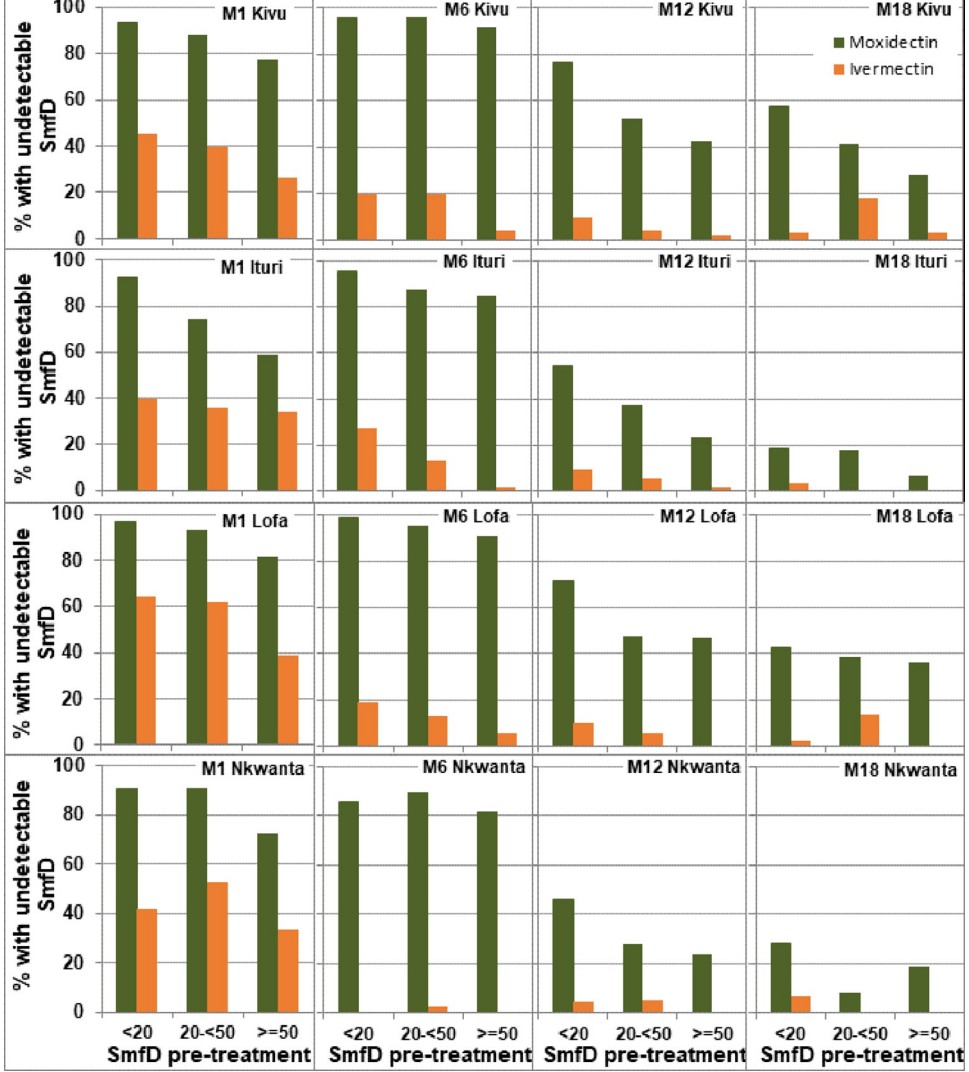

**Fig 4. Percentage of participants with undetectable SmfD by study area, pre-treatment SmfD and post-treatment time.** M1, M6, M12, M18: month 1, 6, 12 18 after treatment (for numbers treated see Fig 3, for numbers with follow up data at each time point, see Tables S4-S8 in **S1 File**).

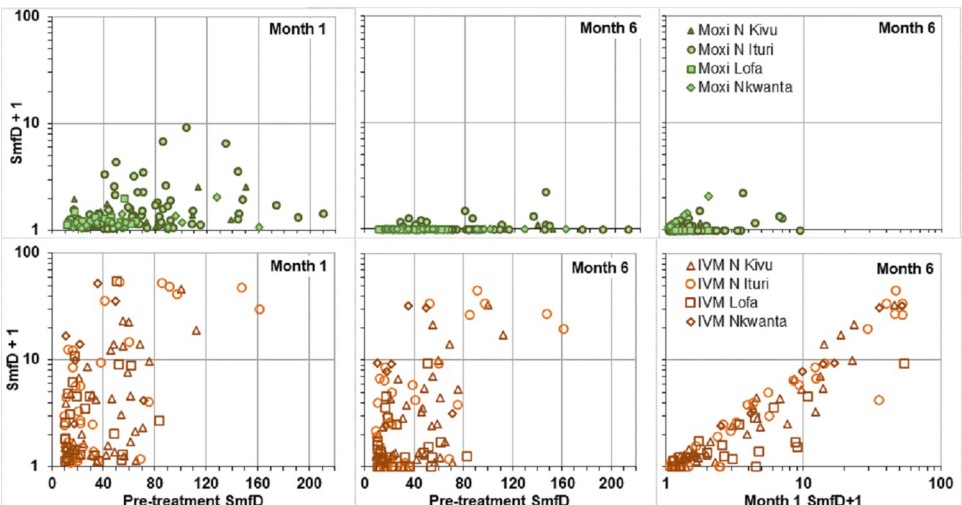

**Fig 5. Skin microfilariae levels pre-treatment, at month 1 and at month 6 among participants who had detectable levels 1 month after treatment.** SmfD: Skin microfilariae density, to allow displaying these data on a logarithmic scale, 1 was added to the SmfD values at Month 1 and 6.

The same trends were observed in the analysis by study area (Fig S4 in **S1 File**). Moxidectin treated participants had significantly higher odds of having UD1-6, 1–12 and 1–18 than ivermectin treated participants (p< 0.008 for each study area) where data was sufficient to allow the comparison in marginal means. The difference in odds for moxidectin and ivermectin treated participants was not significantly different between the different IoI categories (treatment–IoI interaction not significant).

## Inter-individual variability in response to moxidectin and ivermectin

**'Suboptimal microfilariae response' to treatment.** Awadzi et al. considered ≥60% reduction of SmfD from pre-treatment to day 8 as indicating an 'adequate parasite response'. In the first study comparing the effect of moxidectin and ivermectin on SmfD, 3/45 ivermectin treated participants had an SmfD reduction on day 7 or 8 after treatment lower than that and were referred to as 'suboptimal microfilariae responders' [41]. In this study, the first post-

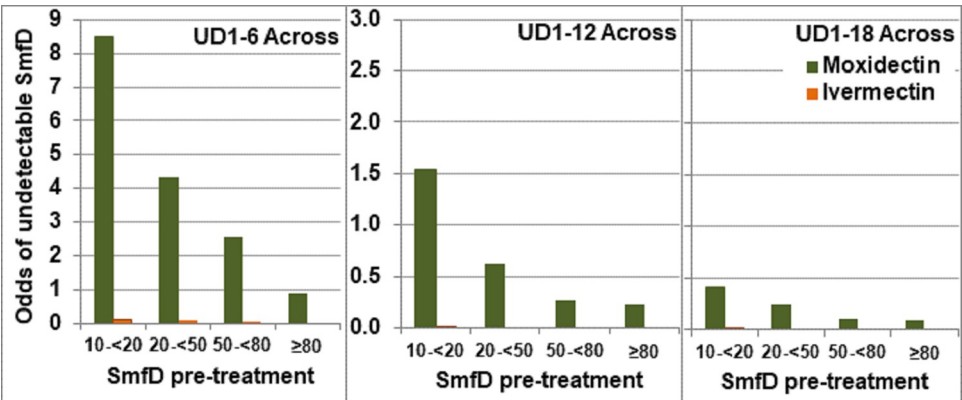

**Fig 6. Odds for undetectable SmfD from 1 to 6, 12 or 18 months by IoI across study areas.** SmfD skin microfilariae density, UD1-6, UD 1–12, UD 1–18 undetectable levels of SmfD at month 1 and 6, at month 1, 6 and 12, and at month 1, 6, 12 and 18, respectively (for number with follow up data at each relevant time point, see Table S11 in **S1 File**).

**Table 4. Percentage of participants with 'suboptimal microfilariae response'.**

| IoI (mf/mg) | DRC Nord Kivu SOMR80 (SOMR90) | | DRC Nord Ituri SOMR80 (SOMR90) | | Liberia Lofa County SOMR80 (SOMR90) | | Ghana Nkwanta district SOMR80 (SOMR90) | |
|---|---|---|---|---|---|---|---|---|
| | Moxi | IVM | Moxi | IVM | Moxi | IVM | Moxi | IVM |
| (Any) ≥10 | 0 (0) | 9.2 (15.6) | 0 (0) | 11.5 (14.7) | 0 (0) | 5.1 (14.1) | 0 (0) | 8.4 (13.3) |
| 10 - <20 | 0 (0) | 5.9 (14.0) | 0 (0) | 18.2 (24.2) | 0 (0) | 7.1 (16.7) | 0 (0) | 12.5 (16.7) |
| ≥20 - <50 | 0 (0) | 11.3 (15.4) | 0 (0) | 11.3 (13.2) | 0 (0) | 2.6 (10.3) | 0 (0) | 5.3 (13.2) |
| ≥ 50 | 0 (0) | 10.2 (18.4) | 0 (0) | 8.5 (11.3) | 0 (0) | 5.6 (16.7) | 0 (0) | 9.5 (9.5) |

IVM ivermectin, Moxi moxidectin, SOMR80: suboptimal microfilariae response based on ≤ 80% SmfD reduction from pre-treatment to month 1 post treatment.

SOMR90 suboptimal microfilariae response based on ≤ 90% skin microfilariae density reduction from pre-treatment to month 1 post treatment

treatment SmfD measurement occurred 1 month after treatment. SmfD decreases from Day 8 to 1–2 months after a single dose of ivermectin [41,45,46]. Therefore, the criterion used here to assess SOMR was reduction of ≤80% in SmfD from pre-treatment to month 1 (SOMR80).

In the moxidectin treatment arm, no individual met the SOMR80 criteria: the % reduction in SmfD from pre-treatment to month 1 ranged from 91.9–100% in Nord Kivu, 94.2–100% in Nord Ituri, 96.9% - 100% in Lofa County, 99.0–100% in the Nkwanta District. All 42 individuals with <99% SmfD reduction at month 1 had >99% SmfD reduction at month 6 (Table 4).

In the ivermectin arm, the percentage reduction in SmfD from pre-treatment to month 1 ranged from 51.0%– 100% in Nord Kivu, 2%– 100% in Nord Ituri, -4.7%– 100% in Lofa County and -49.6%– 100% in Nkwanta District. Across study areas, 8.9% of ivermectin treated met the SOMR80 criteria. Fig 7 shows the SmfD at each post treatment time point for the 44 individuals meeting the SOMR80 criteria. Among these, 21 also met the 'suboptimal response' (SOR) criterion (see below), including 4/14 in Nord Kivu, 10/18 in Nord Ituri, 1/5 in Lofa Country and 6/7 in Nkwanta District. The data do not suggest a dependency of the percentage of SOMR individuals on IoI or study area. This is also not the case when stricter criteria for SOMR are used, i.e. reduction of ≤90% in SmfD from pre-treatment to month 1 (SOMR90) (Table 5).

**Minimum SmfD in participants with detectable SmfD at all post-treatment measurements to Month 12.** Among the 965 moxidectin and 491 ivermectin treated participants with Month 1 as well as Month 6 and/or 12 SmfD measurement, 19 (1.9%) and 257 (52.3%) of participants had detectable SmfD at all evaluated time points (range after moxidectin treatment 0.1–1.2 mf/mg, range after ivermectin treatment 0.1–36.8 mf/mg), respectively. The % of participants who did not achieve undetectable SmfD after treatment was highest among participants in the IoI ≥50 mf/mg category (Table 6). There was no correlation between pre-treatment SmfD and the minimum detectable post-treatment SmfD among either moxidectin or ivermectin treated individuals (Fig 8).

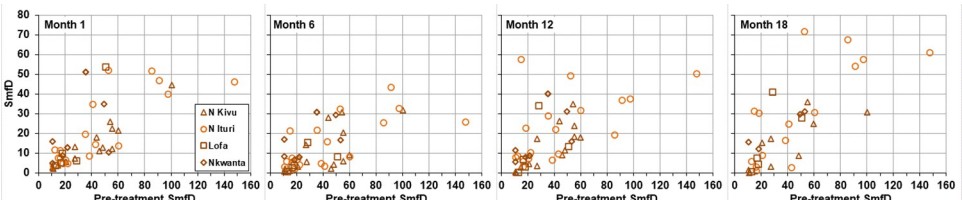

**Fig 7. SmfD 1, 6, 12 and 18 month post-treatment in participants with 'suboptimal microfilariae response' to ivermectin as indicated by ≤80% reduction of skin microfilariae levels from pre-treatment to 1 month after treatment'. SmfD: skin microfilariae density.**

**Table 5. Skin microfilariae density 1 month after ivermectin among participants without and with 'suboptimal microfilariae response'.**

| IoI (mf/mg) | DRC Nord Kivu | | DRC Nord Ituri | | Liberia Lofa County | | Ghana Nkwanta district | |
|---|---|---|---|---|---|---|---|---|
| | OMR | SOMR80 | OMR | SOMR80 | OMR | SOMR80 | OMR | SOMR80 |
| | AM±SD | AM±SD | AM±SD | AM±SD | AM±SD | AM±SD | AM±SD | AM±SD |
| (Any) ≥10 | 0.9±2.2 | 14.7±11.5 | 0.8±2.6 | 21.3±18.1 | 0.6±1.4 | 15.7±21.3 | 0.6±1.3 | 19.9±16.9 |
| 10 - <20 | 0.4±0.6 | 3.0±0.8 | 0.3±0.5 | 7.6±3.3 | 0.3±0.6 | 6.2±3.1 | 0.3±0.5 | 9.9±5.5 |
| ≥20 - <50 | 0.6±1.0 | 11.5±4.4 | 0.5±0.7 | 14.7±11.2 | 0.5±1.0 | 6.0 | 0.7±1.6 | 32.1±27.2 |
| ≥ 50 | 1.9±3.6 | 25.4±11.9 | 1.3±3.8 | 41.5±14.5 | 1.3±2.6 | 53.6 | 0.8±1.4 | 22.7±17.1 |

OMR: Optimal microfilariae response to ivermectin based on >80% reduction of skin microfilariae density from pre-treatment by Month 1, SOMR80 suboptimal microfilariae response to ivermectin based on ≤80% reduction of SmfD from pretreatment by Month 1, AM arithmetic mean SmfD at Month 1, IVM ivermectin, Moxi moxidectin, SD standard deviation of arithmetic mean SmfD at Month1

**'Suboptimal response' to treatment.** A number of criteria have been developed for 'suboptimal response' (SOR) to ivermectin, i.e. individuals in whom the initial decrease in SmfD is followed by an increase larger than expected or considered adequate or typical (for overview of criteria and references see supplementary information in [42]). Using the SOR criterion of SmfD at month 12 post treatment of >40% of the pre-treatment SmfD proposed during the first investigation into SOR [45,47], the percentage of participants with SOR to moxidectin (10/947, 1.1%) was substantially lower than the percentage of participants with SOR to ivermectin (88/480, 18.3%). Among those with SOR to ivermectin, 21 also met the SOMR80 criterion, including 4/14 in Nord Kivu, 10/18 in Nord Ituri, 1/5 in Lofa Country and 6/7 in Nkwanta District. For both drugs, the percentage of participants with SOR differed between study areas but did not increase with IoI category (Table 7). The independence of SOR of IoI is further supported by the across study area analysis of individuals with IoI ≥80: Among the 83 moxidectin treated and 52 ivermecin treated individuals with IoI ≥80 and month 12 data, 0.0% and 21.2%, respectively, met the SOR criterion. Fig 9 shows for each participant meeting the SOR criterion the SmfD at each measurement by pre-treatment SmfD.

## Discussion

Moxidectin treatment induced a maximum SmfD reduction from pre-treatment by 100% in 98.1% of the 978 treated participants and by 99.1%-99.9% in the remaining 19 participants. In conjunction with the minimum post-treatment SmfD of these 19 individuals (0.1–1.2 mf/mg), the data did not suggest any dependency of SmfD reduction on IoI (Fig 8) or study area. Among the 494 ivermectin treated individuals, the by-study area analysis identified 5.1% - 11.5% SOMR80 (individuals with a SmfD 1 month after treatment reduced by ≤80% from

**Table 6. Participants with detectable SmfD and SmfD data at 1 and 6 and/or 12 months post-treatment.**

| | DRC Nord Kivu | | DRC Nord Ituri | | Liberia Lofa County | | Ghana Nkwanta district | |
|---|---|---|---|---|---|---|---|---|
| IoI | Moxi | IVM | Moxi | IVM | Moxi | IVM | Moxi | IVM |
| (mf/mg) | n (%) | n (%) | n (%) | n (%) | n (%) | n (%) | n (%) | n (%) |
| (Any) ≥10 | 1 (0.3) | 88 (57.9) | 13 (4.2) | 91 (58) | 0 | 33 (33.3) | 5 (3.2) | 45 (54.2) |
| 10 - <20 | 0 | 24 (27.4) | 0 | 16 (48.5) | 0 | 12 (28.6) | 1 (2.4) | 13 (54.2) |
| ≥20 - <50 | 1 (0.7) | 29 (56.9) | 7 (4.9) | 29 (54.7) | 0 | 11 (28.2) | 2 (2.7) | 18 (47.4) |
| ≥ 50 | 0 | 35 (70.0) | 6 (5.7) | 46 (64.8) | 0 | 10 (55.6) | 2 (5.1) | 14 (66.7) |

IVM ivermectin, Moxi moxidectin, SmfD Skin microfilariae density

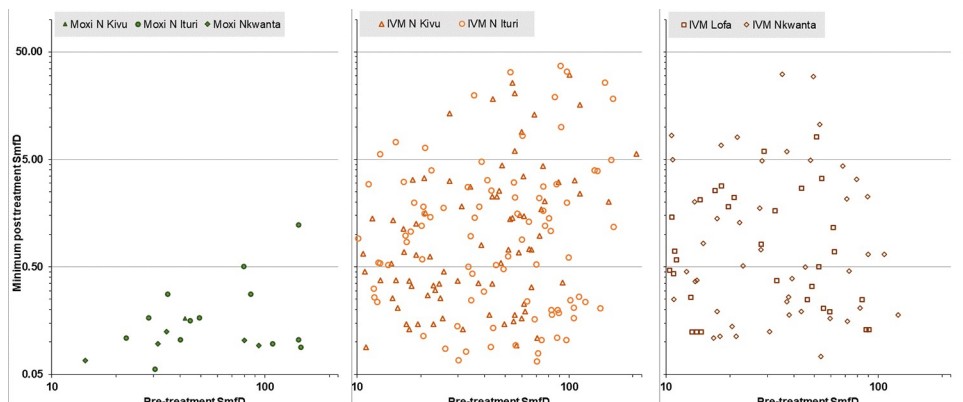

**Fig 8. Minimum SmfD for participants with detectable SmfD at all post-treatment evaluations.** Left: moxidectin treated participants, center: ivermectin treated participants from DRC, right: ivermectin treated participants from Lofa country and Nkwanta North, IVM ivermectin, Moxi moxidectin, SmfD Skin microfilariae density.

pre-treatment SmfD). Between 31.3% and 58.0% of ivermectin treated individuals with Month 1 and Month 6 and/or Month 12 measurements had detectable SmfD at all post-treatment time points. The data did not suggest a dependency of the extent of initial SmfD reduction on IoI or study area and the SmfD in most of these individuals were too high to be attributed to the sensitivity of the skin snip method [48] (Table 5 and Fig 8).

Despite the percentages of ivermectin treated individuals who met the SOMR80 criterion or did not have undetectable SmfD at any post-treatment time, the reduction from pre-treatment to Month 1 in the ivermectin treatment group ranged from 97.4–97.9%. This is similar to the 98–99% reduction at 1–2 months post treatment derived from meta-analysis of the aggregate data from 26 clinical and community single ivermectin dose studies conducted in ivermectin-naïve individuals in Liberia, Mali, Ghana, Cameroon, Senegal, Togo, Ethiopia, Sierra Leone, Cote d'Ivoire and Guatemala [46] and the ≈98% reduction 15 days after treatment among ivermectin-naïve and ivermectin multi-treated individuals from the Nkam and Mbam valley in Cameroon [49]. Given that the skin microfilariae are the reservoir for parasite transmission, including inter-individual heterogeneity in SmfD reduction may be important when modelling the effect of CDTI on time to parasite elimination [50,51].

The possibility of long term use of ivermectin resulting in *O. volvulus* developing resistance has been considered since 1990, i.e. even before initiation of large-scale ivermectin distribution [13,52–56]. In 1997, 10 years after the introduction of large-scale distribution of ivermectin in the Pru and Lower Black Volta river basins in Ghana, field surveys identified individuals with

**Table 7. Percentage of participants with 'suboptimal response' to moxidectin or ivermectin.**

|  | DRC Nord Kivu | | DRC Nord Ituri | | Liberia Lofa County | | Ghana Nkwanta district | |
|---|---|---|---|---|---|---|---|---|
| IoI | Moxi | IVM | Moxi | IVM | Moxi | IVM | Moxi | IVM |
| (mf/mg) | % | % | % | % | % | % | % | % |
| (Any) ≥10 | 0 | 12.1 | 0.3 | 23.7 | 1.6 | 10.8 | 3.9 | 28.0 |
| 10 - <20 | 0 | 3.9 | 0 | 33.3 | 2.6 | 12.2 | 2.4 | 34.8 |
| ≥20 - <50 | 0 | 14.0 | 0.7 | 20.8 | 1.3 | 8.3 | 5.3 | 23.7 |
| ≥ 50 | 0 | 18.8 | 0 | 21.4 | 0.0 | 12.5 | 2.6 | 28.6 |

IoI–Intensity of infection quantitated as microfilariae/mg skin pre-treatment. Suboptimal response defined as SmfD 12 months post treatment >40% of pre-treatment SmfD. IVM ivermectin, Moxi moxidectin, For the number of participants with data at month 12 see Tables S5-S8 in **S1 File**.

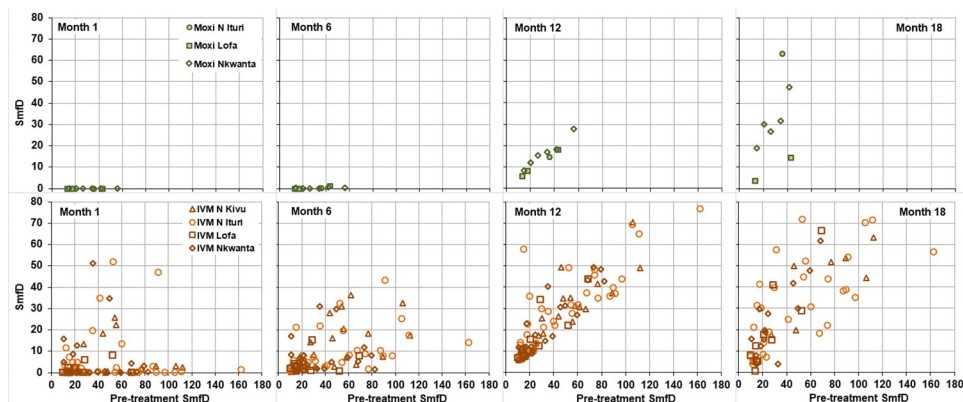

**Fig 9. SmfD at Month 1, 6, 12 and 18 vs pre-treatment SmfD for the 10 moxidectin and 88 ivermectin treated participants with 'suboptimal response' participants with 'suboptimal response'.** IVM ivermectin, Moxi moxidectin, SmfD Skin microfilariae density.

'persistent, significant microfilaridermia'. This motivated the first study examining the presence and factors potentially contributing to a putatively reduced efficacy of ivermectin, initiated in 2001. The publication by Awadzi et al. introduced the term 'suboptimal' for SmfD 12 months post treatment larger than considered 'adequate', attributed to tolerance or resistance of the adult female worm to ivermectin's temporary embryostatic effect [45,47]. Further studies were conducted in Ghana [57–59] and Cameroon [60]. Our study allows us to complement the available body of evidence with data on the prevalence of 'suboptimal response' in ivermectin-naïve individuals in our study areas, including an area in Ghana not included in previous studies.

In contrast to the initial SmfD reduction, the repopulation of the skin with microfilariae depended on IoI. This is consistent with the fact that, apart from reflecting the host immune response [61,62], the IoI in an ivermectin-naïve individual reflects the number of reproductively active macrofilariae. This is also consistent with our previous analysis which considered only the two IoI categories according to which individuals had been stratified for randomization (10–20 mf/mg, ≥20) [42] and analyses of data obtained after the first ivermectin treatment of individuals from Cameroon [60]. The dependency was modest after moxidectin but pronounced after ivermectin treatment (Figs 2 and 3). This difference resulted in individuals with higher IoI benefitting relatively more from moxidectin treatment than individuals with lower IoI as indicated by across-study treatment differences 12 months post treatment of 74.1%, 84.2%, 90.0% and 95.4% among those with IoI of 10-<20, ≥20-<50, ≥50-<80, ≥80.

The by-study area analysis showed that repopulation of the skin with microfilariae, evident at month 6 in the ivermectin but only at month 12 in the moxidectin group, was more extensive among individuals in Nord Ituri and Nkwanta North than individuals in Nord Kivu and Lofa Country (Fig 1). This difference was not driven exclusively by differences in IoI distribution among the participants in the different study areas (Fig 3). The percentages of participants with SOR to ivermectin were higher in Nord Ituri and Nkwanta North District (23.7% and 28%, resp.) than in Nord Kivu and Lofa County (12.1% and 10.8%, resp.). The percentage of SOR was not consistently higher among individuals with IoI ≥50 than among those with IoI 10–20 and thus not driven by the IoI.

The study by Awadzi et al. excluded pharmacokinetic reasons for higher than considered adequate skin repopulation with microfilariae. It concluded that individuals with SmfD 3 and 12 months after treatment exceeding 6% and 40% of pre-ivermectin treatment SmfD

(calculated per 4 mg skin based on four snips), respectively, and embryonic stages in the uteri of macrofilariae 90 days after treatment harbor parasites who respond as expected to ivermectin's 'microfilaricidal effect' but are non-responsive or suboptimally responsive to ivermectin's 'embryostatic effect' [45,47]. Ivermectin's 'embryostatic effect' results in skin repopulation with microfilariae starting weeks after ivermectin has been cleared [63]. Ivermectin has a half life of <1 day [64]. Investigation in Cameroon of skin repopulation rates in ivermectin-naïve individuals from the Nkam valley where CDTI had not yet been initiated and multi-treated individuals from the Mbam Valley with ongoing CDTI showed that skin repopulation rates were higher in multi-treated than ivermectin-naïve individuals to 80 days but comparable to 180 days after ivermectin treatment [49]. The investigators of further studies in the Lower Black Volta, Pru, and Daka river basins suggested that detectable levels of skin microfilariae (based on two snips) 90 days after treatment, SmfD (calculated per snip) 12 months after treatment exceeding 100% of pre-treatment SmfD and/or presence of embryonic stages of microfilariae in the uteri of macrofilariae 90 days after treatment may indicate emerging parasite resistance to ivermectin's embryostatic effect [57,58].

Provided that microfilariae produced by SOR *O. volvulus* have the same fitness as microfilariae from *O. volvulus* responding 'normally / as expected' to the embryostatic effect of ivermectin (i.e. the same probability of being ingested by the vector, developing into infective larvae, being transmitted and developing into reproductively capable macrofilariae if transmitted), earlier skin repopulation with the microfilariae from SOR than 'normally responding' parasites could result in increasing prevalence of SOR parasites if their appearance in the skin occurs while vectors are abundant. This emphasizes the importance of timing and frequency of CDTI relative to transmission seasons. Optimization of CDTI timing was one of the recommendations emerging from the APOC consultations on 'Strategic Options and Alternative Treatment Strategies for Accelerating Onchocerciasis Elimination in Africa [22]. Such considerations would be less important if moxidectin rather than ivermectin was used for onchocerciasis control and elimination [65].

Our data on prevalence of SOR to ivermectin also highlight the need for longitudinal analysis of available data to determine whether or not SOR prevalence is increasing with CDTI duration. Should this be the case, tools suitable for large scale monitoring of SOR prevalence would be needed to guide national program decisions. Development of tools based on genetic correlates of SOR is one approach [66], but the need for phenotypic characterization via serial skin snipping or embryogramms remains a laborious and costly challenge [67].

CDTI had not yet been implemented in the areas where the study was conducted. Consequently, ivermectin selection pressure may not be the explanation for the differences in skin repopulation between the study areas that cannot be attributed to differences in IoI distribution. It takes 10–15 months for a transmitted larvae to mature into a microfilariae-releasing macrofilaria [1]. Therefore, it is unlikely that differences in endemicity and new infections between study areas are responsible for the differences in skin microfilariae repopulation to month 12. Consideration thus needs to be given to the possibility that 'ivermectin-naïve' *O. volvulus* populations in different areas differ in their susceptibility to ivermectin (and to a much lower degree to moxidectin). To date, this possibility has not yet been considered in analyses of data from different geographic areas [46,68]. The hypothesis that parasite populations in different geographic areas have different susceptibility to ivermectin and different inter-individual variability of this susceptibility is compatible with *O. volvulus* biology. Bottlenecks within the life cycle of *O. volvulus* severely restrict the fraction of the progeny of one generation which contribute to the next generation and genetic drift could result in different genetic make up of different parasite populations. Genome-wide association analyses of the *O. volvulus* nuclear genome of phenotypically characterized macrofilariae from Ghana [57,58]

and Cameroon [49,69,70] suggest significant genetic differentiation between the parasite populations investigated [66].

While our results are consistent with the hypothesis of different ivermectin susceptibility of different ivermectin-naïve parasite populations, they cannot be considered as supporting this hypothesis without further investigation: while the areas where we recruited participants were CDTI-naïve, areas not far away (e.g. for Nkwanta district other villages within the Oti river basin, in the Pru River basins and across the border in Togo) had been benefitting from CDTI for many years. The parasites in our study participants could belong to the same parasite population as the parasites in individuals in nearby areas which have been undergoing CDTI for many years. The population genetic structure of parasites obtained in the SOR studies in Ghana [57,58] suggests that parasites in individuals in the Pru, Daka and Black Volta/Tombe river basins belong to the same transmission zone in which parasites interbreed, enabled through vector or people movement ([67]). Without genetic analysis of the parasites from the Kpasa subdistrict in Nkwanta North it cannot be excluded that they belong to that same interbreeding parasite population. If that is the case and if CDTI selects for SOR, this selection in the Pru, Daka and Black Volta/Tombe river basins could be reflected in the parasite population in our study participants. Long range transmission of parasites through infected/infective vector movement was a challenge for the Onchocerciasis Control Programme in West Africa, resulting in extensions of the original programme area [71–74]. Recently long range vector movement has been implicated in renewed transmission of *O. volvulus* in the Comoé valley of Burkina Faso, although movement of infected people could not be excluded [75,76].

The differences in efficacy of ivermectin and moxidectin in terms of the initial decrease in SmfD and in terms of maintaining undetectable or low SmfD through 18 months post treatment resulted in higher odds of moxidectin compared to ivermectin treated participants having undetectable SmfD at month 1–6, month 1–12 and month 1–18 in each study area and each IoI category. Given that the skin microfilariae are the reservoir of transmission, this confirms our conclusion from the across study area analysis that moxidectin treatment may be particularly advantageous for achieving elimination of *O. volvulus* transmission in areas where infection prevalence remains high despite long-term CDTI, where transmission seasons are long or have two peaks, where elimination programmes face operational barriers to implementing at least annual CDTI and where SOR to ivermectin has been observed [42]. This conclusion is supported by the results of modelling the effect of annual or biannual community directed treatment with moxidectin compared to annual or biannual CDTI [77].

Data beyond those which supported the US FDA approval are needed to inform the systematic reviews and evaluation of available data that will be done by WHO [78] and countries to decide whether to include moxidectin in WHO guidelines and country policies for onchocerciasis elimination. A study comparing the effect of 3 annual and 5 biannual treatments with 8 mg moxidectin or 150 µg/kg ivermectin is currently ongoing in Ituri, DRC (https://www.clinicaltrials.gov/ct2/show/NCT03876262). The study will provide comparative data on the safety after repeated treatments, the relative efficacy of cumulative moxidectin and ivermectin treatments overall and by pre-treatment SmfD and on whether variability in inter-individual response to either drug is random or systematic (i.e. individuals with low response to the first treatment will have a low response to subsequent treatments). An ivermectin-controlled single dose study is enrolling individuals with and without detectable SmfD to increase the safety data base (https://www.clinicaltrials.gov/ct2/show/NCT04311671). A paediatric study is identifying a moxidectin dose which will result in exposures in 4–11 year olds comparable to those obtained after treatment with 8 mg moxidectin in ≥12 year old individuals. The protocols for these studies are available at https://mox4oncho-multimox.net/. The safety data from completed or ongoing studies to evaluate the potential value of moxidectin for treatment and

control/elimination of other parasitic diseases (*Strongyloides stercoralis*, *Opisthorchis viverrini*, *Trichuris trichuria*, *Ascaris lumbricoides* [79–82], lymphatic filariasis (https://www.clinicaltrials.gov/ct2/show/NCT04410406), and scabies (https://www.clinicaltrials.gov/ct2/show/NCT03905265))) could also be included among the evidence evaluated by WHO and countries to inform WHO guidelines and country policies on use of moxidectin in onchocerciasis elimination strategies. A study to compare the safety and efficacy of a single dose of 2 mg moxidectin and 150 µg/kg ivermectin in *Loa loa* microfilaraemic individuals is in preparation (https://www.clinicaltrials.gov/ct2/show/NCT04049851). The results of this study and, if the safety data allow, subsequent studies, will inform WHO guidelines and country policies regarding any use of moxidectin in loiasis co-endemic areas. Finally, development of a paediatric formulation has been initiated for children not able to swallow the current 2 mg tablet whose size was chosen in consideration of children as young as 4 years of age (Project Mini-Mox https://www.edctp.org/call/paediatric-drug-formulations-for-poverty-related-diseases-2019/#).

## Conclusions

The screening data show that each of the four study areas was meso- or hyperendemic. Across and within each of these CDTI-naïve areas, the relative benefit in terms of SmfD to 12 months post treatment for individuals treated with one 8 mg moxidectin dose compared to individuals treated with one standard 150 µg/kg ivermectin dose increased with pre-treatment IoI. The data from one of the ongoing ivermectin-controlled studies of moxidectin will provide further information on this.

The statistically significant higher ivermectin efficacy in Nord Kivu and Lofa County compared to Nord Ituri and the Nkwanta district and moxidectin efficacy in Nord Kivu, Nord Ituri and Lofa County compared to Nkwanta district suggest that the possibility of parasite populations in different areas having different drug susceptibility without prior ivermectin selection pressure needs to be considered and further investigated. Such investigations should take into account both genetic drift and long range transmission of parasites via vector and human movement, i.e. that parasites in the human population investigated may be the progeny of parasites subjected to ivermectin pressure elsewhere.

The substantial inter-individual variability in the extent of initial SmfD reduction and subsequent SmfD increase following ivermectin treatment (and the lower variability following moxidectin treatment) as well as the differences in this variability between our study areas also need further investigation. In light of the fact that high percentages of individuals with suboptimal response were considered an indicator of possible emergence of resistance to ivermectin, our data from CDTI-naïve areas may (with the proviso above) be encouraging. They should motivate systematic analysis of individual participant data from the large number of past ivermectin studies in different areas obtained at different times relative to CDTI deployment to assess the effect of geographic area and duration of CDTI (in the area itself and neighbouring areas, see proviso above) on inter-individual variability and the percentage of individuals with suboptimal response. The outcomes can inform strategies for monitoring and evaluation of long term CDTI, interpretation of the results and appropriate consideration of inter-individual and between area variability in transmission models.

## Supporting information

**S1 File.** Table S1 in S1 File: Sex, age and O. volvulus infection of individuals screened by study area and village. Fig S1 in S1 File: Distribution of skin microfilariae density among those screened by study area and sex. Fig S2 in S1 File: Skin microfilariae density across all

individuals screened by study area and sex. Fig S3 in S1 File: Skin microfilariae density among those screened by study area, sex and age. Table S2 in S1 File: Community infection indicators among screened individuals ≥20 years by study area and village. Table S3 in S1 File: GPS coordinates of participant villages, research center and towns in the vicinity. Table S4 in S1 File: SmfD pre-treatment, 1, 6, 12 and 18 months post treatment across all study areas by intensity of infection pre-treatment. Table S5 in S1 File: SmfD pre-treatment, 1, 6, 12 and 18 months post treatment in Nord-Kivu (DRC) by intensity of infection pre-treatment. Table S6 in S1 File: SmfD pre-treatment, 1, 6, 12 and 18 months post treatment in Nord-Ituri (DRC) by intensity of infection pre-treatment. Table S7 in S1 File: SmfD pre-treatment, 1, 6, 12 and 18 months post treatment in Lofa County, Liberia, by intensity of infection pre-treatment. Table S8 in S1 File: SmfD pre-treatment, 1, 6, 12 and 18 months post treatment in Nkwanta district, Ghana, by intensity of infection pre-treatment. Table S9 in S1 File: Adjusted arithmetic and geometric means and mean differences in SmfD 1, 6, 12 and 18 months post treatment by study area. Table S10 in S1 File: Adjusted SmfD means and mean differences 12 months post-treatment by pre-treatment IoI. Table S11 in S1 File: Participants with undetectable SmfD from Month 1 to 6, 12 or 18 by IoI and area. Fig S4 in S1 File: Odds for UD from 1 to 6, 12 or 18 months by IoI and study area. Table S12 in S1 File: Logistic model derived odds for undetectable levels of skin microfilariae from month 1 sustained to month 6, month 12 or month 18 among participants treated with moxidectin or ivermectin by IoI across and by study area. Table S13 in S1 File: Logistic model derived odds ratios for undetectable levels of skin microfilariae from month 1 sustained to month 6, month 12 or month 18 among participants treated with moxidectin relative to among ivermectin treated participants by IoI by study area.
(PDF)

## Acknowledgments

We acknowledge the contribution to data collection in Liberia of Mr. Mawolo Kpawor, who died in October 2016. We acknowledge Dr Fatorma Bolay who directed study center creation and preparation, community engagement, parasitologist training and management of the Liberian study site (see [42]). He died in March 2021. For others whose contribution to the study we want to acknowledge, see [42].

We are particularly grateful to all study participants for their co-operation.

The authors alone are responsible for the views expressed which do not necessarily represent the views, decisions or policies of the institutions with which the authors are affiliated.

## Author Contributions

**Conceptualization:** Annette C. Kuesel.

**Data curation:** Nicholas O. Opoku, Michel Vaillant, Christine M. Halleux.

**Formal analysis:** Michel Vaillant, Annette C. Kuesel.

**Funding acquisition:** Annette C. Kuesel.

**Investigation:** Didier Bakajika, Eric M. Kanza, Nicholas O. Opoku, Hayford M. Howard, Germain L. Mambandu, Amos Nyathirombo, Maurice M. Nigo, Kambale Kasonia Kennedy, Safari L. Masembe, Mupenzi Mumbere, Kambale Kataliko, Kpehe M. Bolay, Simon K. Attah, George Olipoh, Sampson Asare.

**Methodology:** Annette C. Kuesel.

**Project administration:** Didier Bakajika, Eric M. Kanza, Nicholas O. Opoku, Hayford M. Howard, Christine M. Halleux, Annette C. Kuesel.

**Resources:** Christine M. Halleux, Annette C. Kuesel.

**Software:** Michel Vaillant.

**Supervision:** Didier Bakajika, Eric M. Kanza, Nicholas O. Opoku, Hayford M. Howard, Christine M. Halleux, Annette C. Kuesel.

**Visualization:** Annette C. Kuesel.

**Writing – original draft:** Annette C. Kuesel.

**Writing – review & editing:** Didier Bakajika, Eric M. Kanza, Nicholas O. Opoku, Hayford M. Howard, Germain L. Mambandu, Amos Nyathirombo, Maurice M. Nigo, Kambale Kasonia Kennedy, Safari L. Masembe, Mupenzi Mumbere, Kambale Kataliko, Kpehe M. Bolay, Simon K. Attah, George Olipoh, Sampson Asare, Michel Vaillant, Christine M. Halleux, Annette C. Kuesel.

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
