## [Decision Letter · Decision Letter 0]

31 Jan 2022

Dear Dr. Kuesel,

Thank you very much for submitting your manuscript "Effect of a single dose of 8 mg moxidectin or 150 µg/kg ivermectin on O. volvulus skin microfilariae in a randomized trial: Differences between areas in the Democratic Republic of the Congo, Liberia and Ghana and impact of intensity of infection" for consideration at PLOS Neglected Tropical Diseases. As with all papers reviewed by the journal, your manuscript was reviewed by members of the editorial board and by several independent reviewers. The reviewers appreciated the attention to an important topic. Based on the reviews, we are likely to accept this manuscript for publication, providing that you modify the manuscript according to the review recommendations. 

Sincerely,

Sabine Specht

Associate Editor

Jennifer Keiser

Deputy Editor

Reviewer's Responses to Questions

**Key Review Criteria Required for Acceptance?**

**Methods**

-Are the objectives of the study clearly articulated with a clear testable hypothesis stated?

-Is the study design appropriate to address the stated objectives?

-Is the population clearly described and appropriate for the hypothesis being tested?

-Is the sample size sufficient to ensure adequate power to address the hypothesis being tested?

-Were correct statistical analysis used to support conclusions?

-Are there concerns about ethical or regulatory requirements being met?

Reviewer #1: Methods

-Are the objectives of the study clearly articulated with a clear testable hypothesis stated? Yes, in the manuscript but not in the Abstract nor in the Author summary.

-Is the study design appropriate to address the stated objectives? Yes, to a great extent. Line 249, please define or explain what « adjusted » stands for.

-Is the population clearly described and appropriate for the hypothesis being tested? Yes

-Is the sample size sufficient to ensure adequate power to address the hypothesis being tested? Yes

-Were correct statistical analysis used to support conclusions? Yes

-Are there concerns about ethical or regulatory requirements being met? No

Reviewer #2: The method for me is well done

**Results**

-Does the analysis presented match the analysis plan?

-Are the results clearly and completely presented?

-Are the figures (Tables, Images) of sufficient quality for clarity?

Reviewer #1: Results

-Does the analysis presented match the analysis plan? Yes

-Are the results clearly and completely presented? Yes, mostly. However, the « treatment differences » as provided in the abstract (line 66) are not apprehensible without a clear definition of what that means. Given the complexity of how this criterium was defined, I would recommend using different statistics to provide meaningful results in the abstract.

-Are the figures (Tables, Images) of sufficient quality for clarity? Yes

Reviewer #2: The results are well presented

**Conclusions**

-Are the conclusions supported by the data presented?

-Are the limitations of analysis clearly described?

-Do the authors discuss how these data can be helpful to advance our understanding of the topic under study?

-Is public health relevance addressed?

Reviewer #1: Conclusions

-Are the conclusions supported by the data presented? Yes, but they need to be given in the context of the clinical trial: results after a very first dose of moxidectin or ivermectin. The same comparison after second or subsequent treatments would certainly result in much smaller - if any - differences.

-Are the limitations of analysis clearly described? No, but there is matter to do so (eg: see previous point).

-Do the authors discuss how these data can be helpful to advance our understanding of the topic under study? Yes

-Is public health relevance addressed? Yes it is.

Reviewer #2: The conclusions is in accordance with the objectives of the paper.

**Editorial and Data Presentation Modifications?**

Reviewer #1: Editorial and Data Presentation Modifications?

Use this section for editorial suggestions as well as relatively minor modifications of existing data that would enhance clarity. If the only modifications needed are minor and/or editorial, you may wish to recommend “Minor Revision” or “Accept”. (Limit 20000 Characters) 

Because the paper is quite lengthy. I recommend to move the section stating line 272 and ending line 312 to the supplementary information section. It is only marginally relevant to the subsequent analyses.

Line 233: « for this manuscript » can be omitted 

Line 264: replace « explained » by a more appropriate expression (eg: depending on)

Line 424: UD is defined for undetectable SmfD at line 424 although the expression has been previously used, and is again used in subsequent lines. Reconsider the appropriate use of UD.

Line 515: « typical » should be « atypical »

Line 538: « a maximum SmfD reduction from pre-treatment by 100% » would better read as « mf clearance » of equivalent expression.

Line 638: delete one of the comas

Reviewer #2: Minor comment

SOMR: it is not clear whether this Sub Optimal Microfilariae Response on Sub Optimal Microfilariae Responders (Line 459, page 30 and Table 6 and 7).

OMR: Optimal Microfilariae Response is used in Table 7, but nowhere in the text.

**Summary and General Comments**

Reviewer #1: This is a companion paper to an important already published paper that showed an increased efficacy (and similar safety profile) of a single oral treatment of moxidectin over a single oral dose of ivermectin on Onchocerca volvulus microfilaridermia. The present paper had the specific objective of comparing the efficacy of the two drug across four different study sites. The main conclusion of this trial is that moxidectin seems to be more beneficial when skin mf density is higher. 

I think that the authors’enthusiasm regarding the promising results obtained with moxidectin may guide them to promote moxidectin over ivermectin beyond avaiabe evidence. Indeed, the authors should keep in mind that their results were obtained as part of a clinical trial in which all participants were treated for the first time. Moxidectin, likewise ivermectin, is to be used repeatedly (annually, semi-annually, or possibly at longer intervals if the effects are long standing). The conclusion obtained in this paper may not stand after a second treatment given six months or one year later. Both inter-individual heterogeneity and treatment difference will certainly be reduced during round 2, 3 and so on, compared to round 1 of treatment. This should be addressed in the discussion. 

In addition, for the time being, moxidectin is not recommended for use in children, therefore « CDTM » is not an option for now. I would recommend the authors to temper their conclusions or perspective regarding the long term effect of moxidectin. Maybe it will be much more efficient that ivermectin (allowing for reducing the duration of intervention for similar coverage/compliance figures), maybe the « treatment difference » will only be significant after the initial treatment.

In addition, the authors decided to use sub-optimal response criteria defined for ivermectin to assess possible sub-optimal response with moxidectin. Firstly, relevance of those criteria when not associated with genetic data is arguable, and secondly, other criteria may need to be defined for moxidectin. This should at last be addressed in the discussion. In addition, see my point above on the expected decreasing difference in treatment efficacy after several treatments. 

Specific comments:

Line 86: « Ivermectin may not be sufficiently efficacious to achieve elimination everywhere » is purely speculative. Persistent onchocerciasis is mostly due to a mis-use of ivermectin (low therapeutic coverage, hectic compliance…) not to a lack of efficacy when it is actually used. Onchocerciasis has been eliminated using ivermectin only. Use of moxidectin for community treatment will 

Line 558: « long term » may be omitted from the sentence

Lines 561-562: IoI reflects the number of reproductive adults but also the immunological tolerance of the host towards microfilariae

Reviewer #2: This paper by Bakajika et al, summarize data already published with the huge effect of Moxidectin on onchocerciasis compared to ivermectin after the phase 3 trials across the study sites in DRC, Liberia and Ghana. The main message of the present publication is that the difference between Moxidectin and ivermectin differs according to the study areas. Also, the suboptimal response to ivermectin was described in naïve ivermectin areas. All these information are known from previous data and previous publications. Nevertheless, this paper highlights the priority in the use of Moxidectin and insists on the adding value of this molecule for the elimination of onchocerciasis. Considering the priority and the emergency to eliminate onchocerciasis in Sub-Saharan Africa, this new tool (Moxidectin) should be popularized and widely used for the benefit of the affected populations.

PLOS authors have the option to publish the peer review history of their article (what does this mean?). If published, this will include your full peer review and any attached files.

Reviewer #1: No

Reviewer #2: Yes: Kamgno Joseph

Figure Files:

Data Requirements:

Reproducibility:

References

---

## [Decision Letter · Decision Letter 1]

13 Mar 2022

Dear Dr. Kuesel,

We are pleased to inform you that your manuscript 'Effect of a single dose of 8 mg moxidectin or 150 µg/kg ivermectin on O. volvulus skin microfilariae in a randomized trial: Differences between areas in the Democratic Republic of the Congo, Liberia and Ghana and impact of intensity of infection' has been provisionally accepted for publication in PLOS Neglected Tropical Diseases.

Best regards,

Sabine Specht

Associate Editor

Jennifer Keiser

Deputy Editor

Reviewer's Responses to Questions

**Key Review Criteria Required for Acceptance?**

**Methods**

-Are the objectives of the study clearly articulated with a clear testable hypothesis stated?

-Is the study design appropriate to address the stated objectives?

-Is the population clearly described and appropriate for the hypothesis being tested?

-Is the sample size sufficient to ensure adequate power to address the hypothesis being tested?

-Were correct statistical analysis used to support conclusions?

-Are there concerns about ethical or regulatory requirements being met?

Reviewer #1: Fine

**Results**

-Does the analysis presented match the analysis plan?

-Are the results clearly and completely presented?

-Are the figures (Tables, Images) of sufficient quality for clarity?

Reviewer #1: Fine

**Conclusions**

-Are the conclusions supported by the data presented?

-Are the limitations of analysis clearly described?

-Do the authors discuss how these data can be helpful to advance our understanding of the topic under study?

-Is public health relevance addressed?

Reviewer #1: This new section is very helpful.

**Editorial and Data Presentation Modifications?**

Reviewer #1: (No Response)

**Summary and General Comments**

Reviewer #1: Most confusing or arguable points have been clarified. As a consequence, the revised version reads significantly better than the original version. The new conclusion is very helpful in this regard. Though I suspect a touch of bad faith from the authors regarding some specific issues raised during the previous review (eg: inter-individual heterogeneity after repeated treatments), the present version is, imho, acceptable for publcation by PLOS NTDs.

PLOS authors have the option to publish the peer review history of their article (what does this mean?). If published, this will include your full peer review and any attached files.

Reviewer #1: No

---

## [Editor Report · Acceptance letter]

21 Apr 2022

Dear Dr. Kuesel,

We are delighted to inform you that your manuscript, "Effect of a single dose of 8 mg moxidectin or 150 µg/kg ivermectin on O. volvulus skin microfilariae in a randomized trial: Differences between areas in the Democratic Republic of the Congo, Liberia and Ghana and impact of intensity of infection," has been formally accepted for publication in PLOS Neglected Tropical Diseases.

Best regards,

Shaden Kamhawi

co-Editor-in-Chief

Paul Brindley

co-Editor-in-Chief
